# A Systematic Review of the Most Recent Concepts in Smart Windows Technologies with a Focus on Electrochromics

Marcin Brzezicki 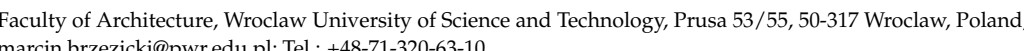

Faculty of Architecture, Wroclaw University of Science and Technology, Prusa 53/55, 50-317 Wroclaw, Poland; marcin.brzezicki@pwr.edu.pl; Tel.: +48-71-320-63-10

**Abstract:** In the context of sustainability and in the face of ambitious goals towards the reduction of $CO_2$ emission, the modification of transparency in architecture becomes an important tool of energy flow management into the building. Windows that dim to stop the energy transfer reduce the cooling load in the building. Recently, however, the latest achievements in the development of electrochromic materials allowed us to integrate some additional—previously unknown—functionalities into EC devices. The purpose of this paper is to provide a systematic review of recent technological innovations in the field of smart windows and present the possibilities of recently established functionalities. This review article outlines recent general progress in electrochromic but concentrates on multicolour and neutral black electrochromism, spectrally selective systems, electrochromic energy storage windows, hybrid EC/TC systems, OLED lighting integrated with the EC device, and EC devices powered by solar cells. The review was based on the most recent publication from the years 2015–2020 recorded in the databases WoS and Scopus.

**Keywords:** smart glass; smart window; electrochromic

## 1. Introduction

Humans need daylight to live a healthy life. Vitamin D is synthesised only in the presence of sunlight. The physiological diurnal cycle of rest and activity is also governed by the day/night cycle. The provision of an adequate amount of daylight indoors is, therefore, a matter of health and well-being. Daylight is allowed through openings into buildings. Windows provide the illumination of rooms where people live. However, a positive impact of contemporary windows in the building is appreciable "as long as they do not cause glare, thermal discomfort, or a loss of privacy" (Hellinga [1]). Together with the visible part of the daylight spectrum, heat (an infrared portion of the spectrum) is also transmitted through glazing into the rooms, resulting in internal temperature build-up. Mechanical removal of excessive heat from a room involves high energy consumption and high environmental costs, such as carbon footprint and exhaustion of non-renewable materials. Buildings are, therefore, of special concern as they are responsible for 30–40% of the world's energy consumption. Cooling, especially, has grown strongly in importance in recent years. Indeed, more energy is required to cool buildings than to heat them [2]. Therefore, in light of the ongoing climate change and global warming, technologies designed to reduce energy demand in the warm season of the year are receiving particular attention because of their significance in achieving the 17 sustainable development goals defined by United Nations [3], with particular attention to goal 11 (sustainable cities and communities) and goal 9 (fostering innovation). Any technology that prevents heat build-up in a building translates into energy savings and reduction of $CO_2$ emissions. In addition, heat loss might contribute significantly to the overall energy balance. Window frames—as an important element of the system—are modelled to evaluate the thermal transmittance to reduce the energy requirement of the whole building [4].

The term "smart window" was coined by Granqvist in 1985 [5]. From the early 1980s, smart glazing has been a rapidly developing innovative technology that is aimed to help

manage energy transfer through the building's envelope, evading unnecessary "cooling and heating of indoor air" [6]. The use of smart glass, which helps to regulate the amount of light (and heat) entering a building, is one of the possible ways to reduce energy consumption in buildings while maintaining an appropriate level of comfort for users. Smart glass greatly influences the building envelope performance in (i) thermal management, (ii) daylight harvesting and regulation, (iii) reduction of glare, (iv) maintenance of views, (v) power capture, and finally (vi) activating the envelope as information display [7]. Some technologies are currently available on the market, although—in light of the many shortcomings of the existing solutions—smart glass is the subject of ongoing "intensive research aimed at improving the technology and its widespread use" [8].

### 1.1. Method and Eligibility Criteria

The data for the review were acquired from international scientific databases (WoS and Scopus—last search 28 January 2021), and the manufacturers' websites (e.g., https://www.saint-gobain-glass.com/products/priva-lite (accessed on 20 August 2021), and other open channels (Google Scholar). The reports analysed were published in English. Candidate search terms ("keywords") were identified by examining the words in the titles and abstracts of the studies included in the previous reviews of smart glass technologies (see Section 1.3. for the detailed description). The search strategy was validated by cross-testing on different databases whether it produces similar results (WoS and Scopus), and the results of this test were positively evaluated. Search strategy for all databases included the papers that featured keywords "smart glass", "smart window" and—after the first refinement was made—also "electrochromic" keyword.

The review was carried out by a single researcher (M.B.). The inclusion algorithm (the procedure to decide which reports were included in the review) consisted of three steps: (i) whether the technology can "dim-on-demand", (ii) whether the technology is reversible, and (iii) the reported technology reached the stage of a small-scale working prototype (big enough to measure the optical transmittance). Certain studies were ineligible to be included in the review as the outcomes were out of the scope of the interest, e.g., because the results were not scalable or the technology did not reach the stage of the working prototype.

Study selection was a multi-stage process in which potentially eligible studies were first identified from screening titles and abstracts. The researcher (M.B.) reviewed titles and abstracts of the 1850 records (published in the years 2015–2020) and, after duplicates removal, 1092 records were screened, from which 145 full-text documents were reviewed, and finally, 105 (each cited) papers were included in the review. Later, a cross-check of the citations was performed, however, no extra articles that fulfilled inclusion criteria were found in these searches (a flow diagram is available see Supplementary Materials). Because of time constraints, only approx. 30% of the titles and abstracts were dually screened; for the rest, a single screening was used. A recent study in epidemiology showed that single abstract screening misses up to 13% of relevant studies [9]. Nevertheless, the author is confident that this methodological limitation would not change the overall conclusions of this review in engineering, as the most recent smart window technologies are well represented in the analysed reports.

Data extraction from the studies was based on the data given by the authors; a small spreadsheet was designed to collect and compare extracted data. The detailed information about the metric is given in Section 2, General Classification and Metrics.

Given the complexity of the report being investigated, the attempt was made to categorise the included reports/papers along four dimensions: (1) passive vs. active technologies, (2) the stimulus that is used to activate the dimming behaviour, and (3) technology scale (micro-, nano-scale). Some of the reports included technologies that had multiple components (e.g., diming on the demand with photovoltaic effect). These reports were categorised according to the main component (the component that the primary authors emphasised, e.g., by stating the keywords).

Results of individual studies were presented in the form of tables, comparing the different results achieved by individual teams of researchers. No statistical synthesis was conducted. The risk of missing or including unclear information from the reported studies is limited, as information concerning a single smart window technology is derived from at least a few sources. This fact increases the credibility of information and reduces the risk of bias. The risk of bias is also reduced by the fact that all analysed reports were peer-reviewed publications. The risk of bias due to missing results is marginal, as the results come from numerous sources included in the paper.

Below-described solutions were studied and systematised to compare the most recent concepts and possible areas of future development in the years 2015–2020.

The main/core scientific method that was used is a desk study; no automation tools were used. The main tool used was a PC with an internet connection.

### 1.2. Rationale behind the Presented Review and Objectives

The current review was considered necessary as previous reviews were either focused on the other area (perspective) or outdated, e.g., reported in the year 2016—previous reviews are discussed in the detail in Section 1.4.

The presented review examines the results of the most recent research and extracts the technologies that are the most promising in the context of the use in sustainable solutions in architecture and building engineering. The objectives of the review were to give a bird's eye view on novel active smart glass technologies (active meaning = dimming on demand) with the focus on electrochromic devices (further addressed as ECDs) which appeared in the years 2015–2020 and were published.

This review aims to present the most recent concepts in active smart glazing that present possible—i.e., available in the future—functionalities, without deeply diving into the issues of material engineering. It is hoped that the materials that are currently in the stage of development would finally result in commercially available products in the building industry.

### 1.3. Originality of the Paper

The purpose of this paper is to give an overview of the most recent solutions in smart glass, with special attention being given to the phenomenon of electrochromism and its use in technical devices. The original aim is to report on the cutting-edge laboratory solutions, which might outline the potential areas of future development. The original approach presented is concentrated on the comparison of the parameters of the devices, mainly visual and NIR modulation. Optical metrics are especially important when it comes to practical application in building engineering and in architecture, from which the author of the paper derives. Metrics are used to show the performance of different solutions that are responsible for reducing the carbon footprint and thus contribute to sustainable development (e.g., NIR modulation that reduces the temperature build-up in the room).

The original method of the presentation of the data in the presented paper derives from the simplified presentation of data that was pioneered by Ke et al. [10] but has been improved. Modulation data (solar, vis and NIR) presented in the paper could be easily compared and are more accessible for a wider audience. Due to the space limitations in the paper, all available parameters related to all the metrics are presented in Supplementary Materials, as well.

A review-type paper has its own rules. As in the case of similar works, in this study, WoS and Scopus databases were used to retrieve the most recent publications related to featured keywords: "smart glass", "smart window", and also "electrochromic". Despite the strict definition of what a review-type paper is, some innovations can be distinguished in the presented work. These are: (i) the use of the PRISMA protocol to process the data (PRISMA flow chart and PRISMA 2020 Checklist are available as Supplementary Materials); (ii) the use of the objective evaluation of presented technologies based on the parameters given by the authors; and (iii) the recognition of the relationship between results of the

laboratory experiments and potential future application of presented technologies. Further, by comparing research results, technologies, and types of innovation, a simplified typology of technologies and featured systems could be established. The results of the review could also be used to find points at which emerging technologies (TRL Proof-of-Concept) and final products (TRL System Adequacy Validated) might overlap in future. The paper also offers a more accurate understanding of how the different research groups can be cross-fertilised to strive for innovation.

### 1.4. Previous Research, State of the Art

The issues of smart glass were recently (2015–2020) extensively studied, mainly in the area of material science, nano-chemistry, and energy simulation. In total, almost 2000 research papers were published on the topic in the years 2015–2020, some of which were review papers. The extensive review of the energy-saving potential of commercially available solutions based on the manufacturer's data was given in 2019 by Tällberg et al. [11] featuring also building energy simulation of the active smart windows. A review of active dynamic windows for buildings was also provided by Casini [12,13], with particular attention given to the main active chromogenic technologies on the market. Ongoing advances in the area of ECD and materials, its modes of operation in different categories alongside existing difficulties, and suggestions to improve performance are summed up by Rai et al. [14]. Ge et al. [15] have provided a less general and more detailed review of one-dimensional $TiO_2$ nanostructured materials for environmental and energy applications. A most recent review of micro shutters for switchable glass was provided by Lamontagne et al. [16]. An interesting lecture summarising the most recent EC technologies was published by Kraft [17] while Granqvist et al. [18], in their review, concentrated on the device longevity and lifetime prediction of EC solutions. In 2019, Aburas et al. provided a review on thermochromic smart window technologies for building applications [19]. Recent reviews were provided by Ke et al. [10] and Wang et al. [20], while Park et al. have concentrated on the improvement in the energy performance of building envelopes incorporating electrochromic windows [21]. In 2021, Feng et al. provided the most recent critical review of fenestration/window system design methods for high-performance buildings [22]. (See Table 1.)

**Table 1.** Previously conducted review research on the topic.

| No. | Team | Year | Focus |
|---|---|---|---|
| 1 | Tällberg et al. | 2019 | energy saving potential of adaptive and controllable smart windows |
| 2 | Casini | 2018 | active dynamic windows for buildings |
| 3 | Rai et al. | 2020 | recent advances in electrochromic devices |
| 4 | Ge et al. | 2016 | a review of one-dimensional $TiO_2$ nanostructured materials |
| 5 | Lamontagne et al. | 2019 | review of micro shutters for switchable glass |
| 6 | Kraft | 2019 | general issues of electrochromism |
| 7 | Granqvist et al. | 2019 | advances in electrochromic coating technology |
| 8 | Aburas | 2019 | thermochromic smart window technologies for building application |
| 9 | Ke at al | 2019 | perspectives on the future of electrochromic |
| 10 | Wang et al. | 2015 | review of switchable materials used in smart windows |
| 11 | Park et al. | 2019 | energy performance of building envelope incorporating electrochromic windows |
| 12 | Feng et al. | 2021 | a critical review of fenestration/window system design methods for high-performance buildings |

## 2. General Classification and Metrics

In general, the term "smart glazing" or "smart window" refers to various technological solutions that change light transmission through the material (usually glass). Optical transparency can be altered qualitatively or quantitatively. A qualitative change is made when a transparent glass pane turns into a translucent one (the amount of light does not significantly change, while the light becomes scattered). A quantitative change is made when the amount of light changes, e.g., the transparent panel is dimmed and blocks the portion of incoming radiation (so-called: coloured/darkened state vs. bleached state). Qualitative technologies are used when privacy is required, while quantitative ones are used to protecting buildings from overheating and users from glare. Although the techniques used to achieve such effects have evolved, they generally fall into one of the two categories mentioned above [8]. The layers of smart materials are usually incorporated into the standard insulating glass unit on the surface No. 2, which means the internal surface of the external pane.

The objective of this work is also to provide the readers with information about the smart window properties. However, it has to be stressed, that the information about the characteristics is difficult to systematise, as the researchers are using different metrics, e.g., the team of Mori [23] is measuring the difference in the increase in in-room temperature when the micro blinds are on and off, without any reports considering optical properties.

*Metrics*

Smart windows, in general, might be characterised by several solar radiation glazing factors, including visible solar transmittance, solar transmittance, ultraviolet solar transmittance solar material protection factor, solar skin protection factor, external visible solar reflectance, internal visible solar reflectance, solar reflectance, solar absorbance, emissivity, solar factor, and colour rendering factor. All these factors are exhaustively discussed by Tällberg et al. [11], based on Jelle [24], who provides a mathematical and physical background for all the measures.

Comparison of these solar quantities for different scientific teams is difficult, as different teams concentrated on different metrics, usually the one that illustrated the result of their research in the most appropriate way.

The term "smart window" is used to describe the whole range of different technologies that can either (i) self-regulate the passage of solar radiation (thermochromic and photochromic) or (ii) regulate the transparency by the application of an external voltage. This change is achieved either by regulation of the absorbance or the reflectance, as addressed in detail by Jelle [24]. Jelle also presents two figures, illustrating the difference between the change in transmittance by a movable reflectance edge and the change in transmittance by a movable absorbance edge. Moreover, technologies exist that feature combined regulation of both absorbance and reflectance. An exemplary diagram—in a different graphic form—is given in Figure 1 for the illustration.

Nguyen et al. state that "most of ( . . . ) smart window research focuses only on modulation within the visible range of the solar radiation ( . . . ) However, since nearly 50% of solar energy comes from IR radiation" [25], dynamic modulation of IR radiation should be also included in the review. To quantify and compare solar characteristics of different glass materials (or the same material in a different energy state), the three most popular metrics are used: (i) solar radiation transmittance $T_{sol}$ or (ii) visible (luminous) transmittance $T_{vis}$ and (iii) near-infrared transmittance $T_{NIR}$. Visible (luminous) transmittance is usually measured at 550–660 nm, while $T_{NIR}$ is usually given for the range of 1000–1600 nm. Many authors also describe the change in light-transmitting properties by describing the "modulation". The modulation level is calculated by subtracting the radiation glazing factors for the same smart window at the "high and low potentials" [24], e.g., according to the formulas below:

$$\Delta T_{sol} = T_{sol(bleached)} - T_{sol(coloured)} \tag{1}$$

$$\Delta T_{vis} = T_{vis(bleached)} - T_{vis(coloured)} \tag{2}$$

$$\Delta T_{NIR} = T_{NIR(bleached)} - T_{NIR(coloured)} \tag{3}$$

For the sake of simplicity, the modulation will be used thought the paper to characterise the presented solutions and technologies, however, three different values of the modulation will be given of $\Delta T_{sol}$, $\Delta T_{vis}$, and $\Delta T_{NIR}$ (Equations (1) and (3)) as given by the different authors in their papers. However, it must be stated that the authors also provide other metrics, depending on the characteristics of ECD measured. All detailed metrics that were available in the original papers are given in the Supplementary Materials.

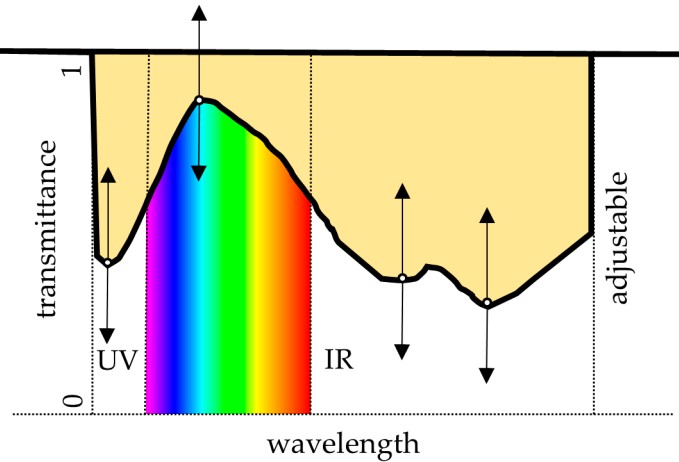

**Figure 1.** Diagram illustrating the change in transmittance in smart glass device by the regulation of absorbance. Diagram by the author modelled on the graphs by Jelle [24].

## 3. Results—Passive Technologies

The light transmission modulation through smart windows can be achieved by many different technologies. These can be divided into passive and active technologies. Passive technologies are those in which the change in the status of a window (e.g., dimming) results from an external stimulus that could not be influenced (e.g., surrounding parameters) without any external regulation. The best examples of passive technologies are glass with photochromic coatings (glass that dims under the influence of sunlight) or thermochromic coatings (glass that dims under the influence of heat). Some authors—e.g., Park et al.—claim that "passive (technology) is typically more suitable for building application as it is automated and its structures are usually simpler" [21]. Although the passive technologies are not in the focus of the presented paper, it is worth reporting that vanadium dioxide ($VO_2$) seems to be a promising alternative for developing thermochromic glazings since its "critical" temperature at which the temperature-dependent properties are changed is at approx. 68 °C, not very far from the usual room temperature [26]. The development of thermochromic—based mainly on vanadium dioxide ($VO_2$)—has led to the design of spectrally selective smart windows which are capable of shielding $\Delta T_{NIR} = 96.2\%$ of the NIR irradiation and transparency modulation of $\Delta T_{vis} = 32.9\%$ [27]. In this device, Lee et al. have used tungsten oxide ($WO_3$)-based EC and vanadium oxide ($VO_2$)-based TC integrated into a single device. Another interesting thermal-based technology is presented by La et al. in a device that can control both the transmittance of solar radiance with the use of thermally responsive material [28]. The team is using the layer of polyampholyte hydrogel (PAH), which is exposing the phase transition in temperatures between 25 and 55 °C (transparency to opacity). In a device, a layer of PAH is heated by an array of electric heaters made of printed elastomeric composite.

Photochromic windows are also actively researched, with the most recent significant results. In 2019, Timmermans et al. [29] reported dual responsive smart widow regulated both by specific wavelengths of light and electrical triggers. The optical response was due to the content of diarylethene dye incorporated in liquid crystals. Enhanced colouration/bleaching photochromic performance was also reported in 2019 by the team of

Li et al. [30]. The device was based on tungsten trioxide ($WO_3$) that constituted a composite matrix with polyurethane (PU) and polyvinyl pyrrolidone (PVP).

Photo- and thermochromic smart windows are promising technologies, but unfortunately, they do not actively influence the transmission modulation of smart windows. For example, the photochromic glass will be dimmed on a sunny winter day when the greenhouse effect is desired, especially in passive buildings. Similarly, the thermochromic glass will be dimmed on warm days, even if we want to keep the light transmission unchanged, for the reason of, e.g., keeping the proper level of daylight in the room.

An important element of passive technologies is also Phase Change Materials, which react to heat by changing the state from solid (light-scattering) to liquid (light-transmitting). It is important to remember that PCM offers control over the quality of light but is not possible to control on-demand. It was also recently reported by Chou et al. that a passive smart window was proposed with the use of thermotropic hydrogel containing graphene oxide, which changes the state from opaque to transparent under the influence of solar radiation. In this solution, the hydrogel can effectively convert the "photoenergy of sunlight into thermal energy and cause the smart glass to reach an opaque state owing to the increased temperature of the hydrogel heated by solar light" [31]. In 2019, Kim et al. [32] recently presented a device featuring a phase transition of the thermosensitive hydrogel that exhibited optical transition from transparent to opaque state. The phase of the gel was controlled by the film of nanopatterned silver, which effectively generated the heat by the Joule-heating mechanism. Table 2 features the schematic illustrating all the described technologies:

**Table 2.** Typological diagram illustrating the described technologies. Diagram by the author.

| Type | Stimulus | Technology | Featured Systems |
|---|---|---|---|
| Passive technologies: | Heat—Thermochromic | | |
| | Light—Photochromic | | |
| | Heat—Phase Change Materials | | |
| Active technologies: | Gas—Gasochromic | | |
| | Fluid—Optofluidic glass | | |
| | Electrical current: | Microsystems | |
| | | Microwrinkled Nanometric Films | |
| | | Polymer dispersed liquid crystal | |
| | | Suspended particle devices (SPD) | |
| | | Electrochromic: | Multicolour EC |
| | | | Neutral black electrochromism |
| | | | Spectrally selective systems NIR/VIS |
| | | | Electrochromic energy storage window |
| | | | Hybrid EC/TC solutions |
| | | | EC devices powered by solar cells |
| | | | Nanostructures |

## 4. Results—Active Technologies

Active solutions are implemented using several groups of different technologies. They can be divided into several groups, depending on the stimulus that causes the transmission modulation of the smart window.

### 4.1. Gas

Gasochromic windows (GC) can change their transmittance in the presence of gas—usually diluted hydrogen with some addition of argon—that induces the reduction reaction of the gasochromic layer, resulting in colouring. Two main substances are used: (1) a layer

of tungsten trioxide ($WO_3$) covered by a very thin layer of silver or (2) magnesium yttrium (Mg-Y) alloy. In the first technology, Wittwer et al. produced porous, columnar film of $WO_3$ by sputtering, and used a low concentration of $H_2$ to change the colour of gasochromic film. The reverse reaction is obtained with the use of $O_2$, which bleaches the film to the original transparent state [33]. In the second technology, Liang et al. produced a device with a $WO_3$ layer, which—after being exposed to diluted $H_2$ at room temperature—is hydrogenated, which leads to the blue tinting in approx. 5 s (coloured state). The dehydrogenation process is initiated by the use of diluted $O_2$, which leads to an increase in transmittance (bleaching) [34]. Additionally, magnesium yttrium (Mg-Y) alloys could be also used in the manufacturing of switchable mirrors. The energy efficiency of the latter technology in the building is currently discussed as the gasochromic Mg-Y layer does effectively block the heat, but the corresponding lower solar transmittance reduces daylight availability and the energy consumption for artificial lighting increases.

### 4.2. Fluid

Optofluidic glass is based on the principle of refractive index matching. The optofluidic window features two layers of transparent material (one of which is roughened/pattered from the inside) and an air cavity between. A roughened surface causes the light rays to reflect and scatter, reducing the light transmittance. When the fluid of specific refractive index matching with the index of the material with roughened/patterned surface is introduced into the cavity, light transmittance is increased. Optofluidic smart windows suffer from many potential maintenance problems, including leakage and the influence of the potential low air temperature (below the freezing point of the liquid), but recently, 3D printing technology allowed for an evident step forward allowing for the manufacture of sealed modules using VeroClear photopolymer. In [35], the team of Wolfe et al. present a novel optofluidic smart glass prototype capable of modulating visible light transmittance ($\Delta T_{vis}$) from 8% to 85% using air (reflective state), water (diffuse transmittance state), and methyl salicylate for specular transmittance. The refractive index of methyl salicylate and photopolymer VeroClear are matched.

Recently, Heiz et al. [36] also presented smart glass that is based on the magneto-active liquid (magnetite nanoparticles in monopropylene glycol) circulating in the cavities/channels parallel to the surface of the glass. The magneto-active liquid is loaded with magnetic nanoparticles, the density of which can be controlled through remote switching in a magnetic particle collector-suspender device in which permanent magnets or electromagnets are used to draw the magnetic nanoparticles from the liquid.

### 4.3. Electrical Current

Smart windows that are controlled by electrical current include a large group of solutions that will be addressed below. Their common feature is that the change in the state of the window requires the flow of electrons (charged ions)—they are electrically activated. This brief review is given below follows the scale of the technology (from macro, through micro- to nano-solutions).

#### 4.3.1. MEMS-Based Microsystems

In general, micro-blinds made of curling electrodes actuated by electrostatic forces belong to the category of microelectromechanical systems [16]. Microelectromechanical systems (MEMS) include microscopic devices, particularly those with moving parts. Microblinds are composed of a trapezoid- or rectangle-shaped curling micro-thin metal blinds on a transparent conductive oxide (TCO). In the absence of voltage, the blinds are curled and light passes through. Once the voltage is applied, the difference of potential is created and the electrostatic force stretches the micro blinds so that light is blocked. Most micro shutters are based on standard microelectronic fabrication processes (e.g., e-beam evaporation, magnetron sputtering, optical lithography). The main advantages of micro shutters are fast (virtually instant) switching time, neutral colouration of the transmitted light, low power

consumption, and stability for UV and temperature. Few institutions currently work on the development of the micro-blinds including the University of Kassel, Germany [37]; Institut National d'Optique, Canada; and University of Tokyo, Japan [23].

Another type of MEMS is micromirror arrays. Each unit consists of the mirror, the hinge, and the steering mechanism. Micromirror glass is composed of millions of electrostatically actuatable micromirrors that can guide and control light dynamically (typical dimensions are $150 \times 400$ mm$^2$). Those systems are used to guide the daylight within the façade, not to block it. Due to the size of the individual mirror, the system is imperceptible for the human eye. The main advantage of the system reported by Hillmer et al. is that the light is reflected, not absorbed, and has low energy consumption, as low as 0.2 mW/m$^2$ [38].

### 4.3.2. Microwrinkled Nanometric Films

As was already mentioned above, the roughness of the transparent surface scatters the light. This phenomenon was exploited in the electrically controlled smart glass device that is using transparent soft media with electrically tuneable surface roughness for transparent-to-translucent switching. The system in a "wrinkled" state scatters the light, while in a "stretched" state becomes transparent. The media used as a membrane in the system is a TiO$_2$ nanometric thin film that is sandwiched between transparent conductive polymers. This system survives 1000 cycles and has a strikingly low power consumption of 0.83 W/m$^2$ [39].

### 4.3.3. PDLC (Polymer Dispersed Liquid Crystal)

A smart window based on the Polymer Dispersed Liquid Crystals (PDLCs) features liquid crystal dispersions in a polymer matrix (simply, microdroplets of liquid crystals encapsulated in a polymer), sandwiched between two transparent conducting electrodes. They scatter light in their OFF state because the molecules liquid crystals are randomly arranged, but become transparent when the voltage is applied in their ON state (when the crystals are ordered) [40]. To remain transparent, PDLC smart windows require the continuous application of an electric field, with an average power consumption of 20 W/m$^2$ as stated by Lampert in [41]. LC molecules embedded in the polymer matrix can be oriented on the demand, thus the transmission could be gradually regulated, as in [42]. This technology is widely used in privacy windows and projection displays because of the fast switching speeds. One of the most widely known commercially available products on the market is Privalite by Saint-Gobain [43].

The latest PDLC technology includes the use of membranes containing liquid crystals with the parameters of an opaque OFF state with a $T_{OFF} = 0.5\%$, and a transparent ON state with a $T_{ON} = 65\%$ (difficult to translate to $\Delta T_{vis}$ as the haze is described) when the system is switched on, as presented by De Filpo et al [44]. Liquid crystals and polymers with other additives and other forms are also studied. Kim et al. solved the dye contamination problems by encapsulating the dye in monodispersed capsules. Using this technology, a fabricated "dye-doped PDLC had a contrast ratio of >120 at 600 nm" [45]. Although PDLC systems are mainly used for privacy purposes, they can also achieve energy savings. Alghamdi et al. recently reported that a system comprising of sensors with an Arduino to control the percentage PDLC glass transparency produced 39% in energy savings compared to the standard systems in a hot climate [46]. Sol et al. recently reported a smart window featuring liquid crystalline luminescent solar concentrator that allows switching the window between three states: "coloured" for increased light absorption, "light" for transparency (5% of haze), and "scattering" for diffuse transmission of light (66% of haze). In the LSC system, luminescent molecules embedded in a polymer absorb light and reemit downshifted spectrum that is channelled by total internal reflection to the edge of the device, where it is collected by PV cells [47].

### 4.3.4. SPD Windows

Suspended particle devices (SPD) smart windows work on a similar principle to PDLC, but instead of liquid crystal, they use a suspension of fine, strongly absorbing particles. In the OFF state, the particles are randomly arranged and block the passage of the light. When the voltage is applied in the ON state, the particles align to let the light through. SPD device requires approx. 5 watts per m$^2$ to remain transparent, as reported by Schwarz [48]. Light transmission values range from about 64–80% in the clear state to 0.5–12% in the dark state [49] as reported by Ghosh. However, it must be noted that due to the number of technological problems, stability, and particle settings, the development of suspended particle devices has been recently slowed. International scientific databases show only less than 10 reports submitted in the years 2015–2020.

The overview of the presented technologies is summarised in Table 3.

**Table 3.** The comparison of the performance of non-electrochromic devices, including the technologies, where the data are available.

| No. | Team | Year | Type | $\Delta T_{sol}$ | $\Delta T_{vis}$ | Remarks |
|-----|------|------|------|---------|---------|---------|
| 1 | Wittwer et al. | 2004 | Active gasochromic | 71% | 72% | Switching from transparent to mirror state |
| 2 | Liang et al. | 2019 | Active gasochromic | 42% | n/a | |
| 3 | Wolfe et al. | 2018 | Optofluidic | n/a | 77% | Clear to foggy |
| 4 | Heiz et al. | 2017 | Magneto-Active Liquid | 95% | n/a | Magnetic particles in liquid |
| 5 | Hillmer et al. | 2018 | Microelectromechanical | n/a | n/a | Micromirrors. The team only measured a temperature build-up in the room. |
| 6 | Mori et al. | 2016 | Electrostatic | n/a | 17% | Micro blinds |
| 7 | Shrestha et al. | 2018 | Microwrinkled TiO$_2$ Films | n/a | 79.2% haze | Transparent to translucent switching |
| 8 | Lampert | 1998 | PDLC | n/a | 40% haze | |
| 9 | Lampert | 2004 | PDLC | 60% | 57% haze | |
| 10 | Murray et al. | 2016 | PDLC | n/a | 25–29% haze | |
| 11 | De Filpo et al. | 2019 | PDLC | n/a | 64% haze | |
| 12 | Sol et al. | 2017 | PDLC | n/a | 61% haze | |
| 13 | Ghosh | 2017 | SPD | 46% | n/a | |

### 4.3.5. ECDs

Electrochromic windows are—according to the survey performed by the author—the leading branch in smart window applications, constituting the majority of search results in international science databases in the years 2015–2020. The technology has been known since the 1960s, when S.K. Deb published important work on the characterisation of molybdenum and tungsten oxide thin films [50]. He had originally observed that some types of metal oxides can change the colour to blue (and brown) due to the reduction reaction and become uncoloured again due to the oxidation reaction.

Electrochromic devices (ECDs), in general, are used for applications ranging from commercialised smart window glasses, goggles, and auto-dimming rear-view mirrors [51]. Recent achievements in electrochromic smart windows technology call for a review study of the most recent concepts that are used to obtain hitherto impossible results, and are studied in detail in the following section.

## 5. Electrochromic Devices

### 5.1. Switching Mechanism

Electrochromism is a reversible chemical phenomenon, where the electrochromic material changes its colour when the voltage is applied. As Kraft writes, the substances, "which change from an uncoloured oxidised state to a coloured reduced state by electrochemical reduction are called cathodic electrochromic, whereas compounds which change from an uncoloured reduced to a coloured oxidised state are called anodic electrochromic compounds" [17]. EC windows operate on the principle of the reversible electrochemical intercalation of positive ions (e.g., $H^+$, $Li^+$, $Na^+$) accompanying the insertion of charge balancing electrons into the multivalent transition metal oxides (e.g., $WO_3$, $NiO$, $IrO$, $MoO_3$, $V_2O_5$) [27]. The basic chemical reaction featuring the most popular cathodic electrochromic compound $WO_3$ transforming from transparent to blue is given below:

$$M x WO3 \rightleftarrows WO_3 + xe^- + xM^+ \quad x \leq 0.3 \tag{4}$$

while nickel oxide (NiO) can be coloured anodically to a brown colour in a reaction of

$$LiNiO \rightleftarrows NiO + Li^+ + e^- \tag{5}$$

However, other transition metal oxides such as $Co_3O_4$, $MoO_3$, $V_2O_5$, $TiO_2$ also exhibit electrochromic properties [52,53]. Prussian blue (iron ferrocyanide)—originally reported by Mortimer [54]—also currently is studied as a material presenting some electrochromic behaviour [55]. Recently, polystyrene sulfonate (PEDOT:PSS) were researched as exhibiting electrochromic properties, as well as presented by Singh [51].

### 5.2. Electrochromic Device Architecture

Electrochromic devices typically consist of five thin layers that are located (sandwiched) between two panes of glass or flexible polyester foil: two external layers of transparent conductive films (usually indium tin oxide, ITO) and the counter electrode (Ni-oxide-based film), electrolyte, and electrochromic electrode in between (W-oxide-based film). The counter electrode is used for ion storage, the electrolyte for conducting ions, and the electrochromic electrode for attracting ions. When the electrical current is applied, the ions stored in the counter electrode (bleached state) migrate through the electrolyte to the electrochromic electrode, resulting in the colouration (coloured state). The mobile ions should be small. Hydrogen protons ($H^+$) or lithium ($Li^+$) ions are commonly used [18]. EC device can be considered as an electrical battery in which the optical absorption is related to its charge; therefore, the crucial component of an ECD is an electrolyte, which can be liquid, gel, or solid, as addressed by Cannavale et al. [56]. The most popular ECD architecture is pictured in Figure 2.

### 5.3. Simulated Energy Performance

Visible light transmission in commercially available electrochromic windows can vary from 3.5% to 62% depending on their operating state [35]. This difference in transmittance translates into energy savings in the building. Picollo et al. report that ECD is effective in reducing heat loads in a cooling dominated climate during the summer season (optimal when the cooling demand is dominant over the heating/lighting demand) [57], while Cannavale et al. report overall yearly energy savings as high as 40 kW h/m$^2$/yr in the hottest climates, assuming clear glazing as a benchmark [58]. Park et al. report a reduction of 8.43% in energy consumption, relative to the reference model when EC device is used, and in a different paper, a reduction of about 11,207 kWh/yr (of 8.89%) for heating/cooling and lighting energy [21]. Generally, Park et al. report the effectiveness of EC technologies particularly in office buildings for which the cooling energy consumption ratio is high. Javad and Navid in 2018 showed a 50% temperature difference reduction between the floor and ceiling, which is achievable with the use of EC windows [59].

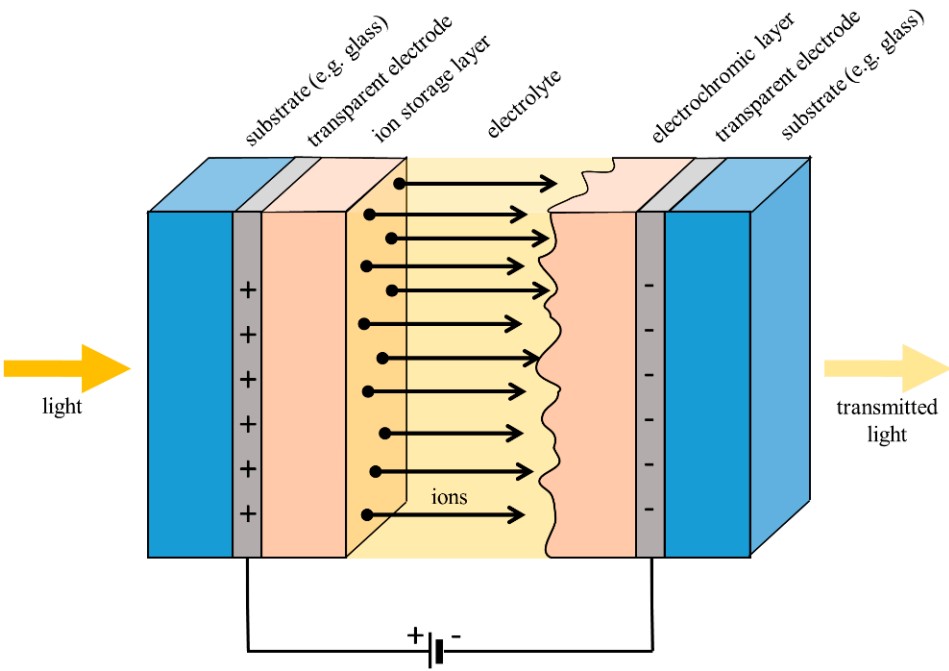

**Figure 2.** The most popular ECD architecture. Diagram by the author.

### 5.4. Most Recent Concepts in EC Smart Windows

As is said above, EC technologies suffer from many problems that are currently being addressed by many research teams tackling the challenges of the EC windows. The concepts of the most recent solutions are briefly discussed below and adequately compared in the tables and the Supplementary Materials.

### 5.4.1. Multicolour EC Solutions

Multicolour solutions are possible thanks to the co-existence of electrochromic materials with different redox potentials. This allows for colour change that is different from the blueish-transparent cycle that is known from tungsten trioxide ($WO_3$). Lee et al. [60] recently reported a multicolour EC device using $Co(OH)_2/Ni(OH)_2$ nanofilms. The nano-thin double layers of $Co(OH)_2$ and $Ni(OH)_2$ were produced using two-step minute electrodepositions. The co-existence of electrochromic materials with different redox potentials allowed for the multicolour change (at −0.2, 0.2, and 0.5 V potentials) among yellow, green, and brown, and those changes were entirely reversible. Some combined polymers with precisely tailored energy gaps have expanded the colour palette/gamut for electrochromic devices. The team of Liu et al. [61] reported an EC device capable of changing from brown, dark brown, purple, to blue using freestanding copolymer P(PVK-co-EDOT) as an electrochromic colouring layer. In the laboratory scale, when potentials were applied between −0.9 V and +0.5 V, the device presented an electrochromic behaviour with a colour reversibly changing from blue to purple. Electrochromic conducting polymers based on the PEDOT structure offer different colours that could be switched electrochemically [62], as reported by Argun et al. The transitions from transparent to magenta, or from blue through green to yellow are possible with the application of potentials of −0.1 V and +0.9 V. The authors also present a summary of different colours that are possible with the use of the different polymers, with an even greater colour palette: yellow-green-blue, or blue-magenta-grey. Futsch et al. [63] report the electrochromic device that is using vanadium oxide ($V_2O_5$) deposited as a micrometric thick film. Depending on the content (different weight percentages of $V_2O_5$ and polymer ink PEDOT), material changes colours from blue to green and orange, depending on the voltage applied (−1.0 V and +1.5 V). The characteristic feature of vanadium oxide is the existence of several oxidation states that offer the possibility of creating multicolour EC windows/displays. The papers on

the electrochromic properties of $V_2O_5$ were recently published by Chu et al. [64] and Mjejri et al. [65].

The main area of the application of multicolour EC windows is seen in architecture as a decoration and possible information display.

### 5.4.2. Neutral Black Electrochromism

Neutral black electrochromism is a Holy Grail of EC devices. A neutral EC device is supposed to absorb all the visible radiation ranging in 400–750 nm in the same degree, and therefore, work as a grey filter. A typical EC device (e.g., based on $WO_3$) is working as a spectrally selective filter altering the colour of the light that is passing through the device. It is usually filtering out specific wavelength ranges. An extensive review of the colour rendering properties of different types of switchable glazing was provided by Aste et al. [66]. Jarosz et al. claim, that black electrochromic devices "grant control over the visible light intensity, indiscriminately of wavelength" [67].

One of the possible solutions to obtain neutral colouration is subtractive colour mixing, usually by setting up a series of different filters that absorb or reflect different complementary wavelength ranges, and, in this way, finally providing the neutral colouration. As it was originally reported by Passerini et al. [68] in 1990, the combination of cathodically blue-colouring electrochromic such as $WO_3$ and the nickel oxide that is an anodically brown-colouring substance might produce an EC device that is changing the colouration from transparent to neutral grey.

However, subtractive colour mixing is regarded as a difficult method in colour creation; therefore, a few other concepts have been tested. These include (i) chemical copolymerisation (different parts of copolymer absorbing defend wavelength ranges), (ii) physical blending (by straightforward mixing of two polymers), (iii) stacking the polymers one on the other, and finally, (iv) by using three-electrode devices, where each electrochromic layer is controlled separately as addressed by the team of Jarosz et al. [67]. Li et al. have reported the neutral EC device based on copolymers, PTTBTPh0.35 and PTTBTTh0.30 that have showed high colour neutrality with CIE 1976 coordinates (L = 37.23; a = −1.59; b = 0.32), when the difference of potentials of −0.4 and +0.8 V was applied [69]. Alesanco et al. [70] recently reported a device that uses asymmetric viologens (ECDs based on a single 1-alkyl-1′-aryl asymmetric viologen) with the simplest device architecture (glass/TCO/EC gel/TCO/glass) that exhibited colourless electrochromism with the parameters of $T_{bleached} \approx 77\%$, and $\Delta T_{vis} = 60\%$.

### 5.4.3. Spectrally Selective Systems

Electrochromic materials alter the visible radiation ranging in 400–750 nm (VIS) but also are capable of reducing NIR (near-infrared, approx. 1000–1500 nm) transmission through the window. As NIR transmission is associated with the heat load filtering, NIR allows for increased thermal comfort while simultaneously decreasing the energy expenditure on cooling. The solutions that are the most intensively sought include those that allow for independent filtering NIR and visible radiation with the use of mainly plasmonic EC materials. Plasmonic nanoparticles are particles whose electron density can couple with electromagnetic radiation of wavelengths that are far larger than the size of the particle, and therefore can exhibit spectrally selective behaviour, as reported by Wu et al. [71].

The system usually works in the four states that are named bright (allowing only the VIS transmittance), warm (VIS + NIR transmitted), cool (only VIS transmitted), and dark (VIS blocked, NIR transmitted). Those states are cycled depending on the voltage applied (usually in the range of −0.5 to +1.2 V, or −3.0 to +3.0). The advances in the field of spectrally selective EC systems were reported by many researchers. Yilmaz et al. [72] have used nanocrystalline ITO (indium-tin-oxide) with high optical contrast of polyaniline (PANI). Laboratory-scale sandwich EC devices were fabricated "using the PANI/ITO nanocomposite film as the active working electrode and a mesoporous $CeO_2$ film (thickness

∼350 nm) as the counter electrode". The synergistic features of PANI/ITO make possible the implementation of a four-state tuneable electrochromic system that permits selectively regulating optical transmittance in the visible and near-infrared range. They achieved an outstanding result of $\Delta T_{NIR}$ = 80%. Another system featuring hybrid multi-layered inverse opal (IO) nanostructure composed of tin ($SnO_2$), titanium ($TiO_2$), and tungsten ($WO_3$) oxides developed by Nguyen et al. [73] was reported in 2019 to modulate up to 63.6% NIR radiation at the wavelength of 1200 nm. Inverse opal (IO) nanostructures are three-dimensionally ordered microporous materials formed through the infiltration of an artificial opal with a material precursor. Earlier in 2019, the same team have reported an IO structure that shows modulation of 70% visible light transparency and 62% NIR blockage at 1200 nm [25]. Other teams use different materials. Transparent amorphous indium zinc oxide (a-IZO) was used by the team of Nunes et al. [74], who reported a selective device that was able to work in two, not four, modes: semi-bright warm mode (VIS + NIR transmitted) and dark cold mode (VIS and NIR blocked) at the modulation $\Delta T_{vis}$ = 50% at 550 nm and $\Delta T_{NIR}$ = 60% at 1000 nm.

Other teams used the different idea of using two electrodes operating in different spectral ranges, one which selectively operates in the NIR and NIR + VIS regimes (usually $WO_3$), and the second which selectively operates in the VIS regime only. The team of Cao et al. [75] used Ta (tantalum) doped titanium oxide ($TiO_2$). The synthesis produces Ta-doped $TiO_2$ NCs as a highly uniform colloidal solution, which is a promising electrode for smart electrochromic windows. The laboratory-scale device exhibited the modulation of $\Delta T_{vis}$ = 86.3% at 550 nm and $\Delta T_{NIR}$ = 81.4% at 1600 nm. In May of 2018, the team of Barawi et al. [76] developed a system featuring four states: fully transparent, VIS blocking, NIR blocking, and VIS and NIR blocking. The device was based on vanadium enriched $TiO_2$. It exploits the peculiar spectro-electrochemical features of colloidal nanocrystals, which exhibit a distinctive electrochromic response at visible wavelengths upon the application of a small cathodic potential. The laboratory-scale device showed a performance of approx. 30% modulation in 550 nm and 70% in 1200 nm. Wu et al. [71] recently reported a device which is capable of shielding 96.2% of the NIR irradiation from 800 to 2500 nm while permitting the acceptable amount of visible light $\Delta T_{vis}$ = 33%, using caesium tungsten bronze ($Cs_xWO_3$). The effect of the optical switching is a direct result of the phase transition of PAM–PNIPAM hydrogel, which in turn is induced by the photothermal effect of $Cs_xWO_3$ under sunlight irradiation. (See Table 4.)

**Table 4.** The comparison of the performance of spectrally selective systems, including the technologies, where the data are available. * Different data for different samples were given.

| No. | Team | Year | Type/Technology | $\Delta T_{vis}$/$\Delta T_{NIR}$ | Remarks |
|---|---|---|---|---|---|
| 1 | Yilmaz et al. | 2020 | nanocrystalline ITO | 44%/77% | Cool/Warm/Dark states |
| 2 | Nguyen et al. | 2019 | ($SnO_2$), ($TiO_2$), ($WO_3$) | 21%/64% | Vis/NIR |
| 3 | Nguyen et al. | 2019 | opal (IO) nanostructures | 12%/57% * | Vis/NIR |
| 4 | Nunes et al. | 2019 | a-IZO | 45%/57% | NIR at 1000 nm |
| 5 | Cao et al. | 2018 | Ta (tantalum) doped titanium oxide ($TiO_2$) | 86%/81% | NIR at 1600 nm |
| 6 | Barawi et al. | 2018 | vanadium enriched $TiO_2$ | 30%/70% | NIR at 1500 nm |
| 7 | Wu et al. | 2018 | caesium tungsten bronze ($Cs_xWO_3$) | 33%/96.2% | NIR at 800 to 2500 nm |

### 5.4.4. Electrochromic Energy Storage Window (EESD)

Standard EC devices (windows) operate by consuming electrical energy to stimulate the chemical reaction of reduction or oxidation with negligible energy storage ability. By the analogy of standard EC window to the battery—with the use of proper materials—the device shows the electrical phenomenon of pseudocapacitance, which is the storage

of electricity in an electrochemical capacitor. The energy is stored by electron charge transfer between electrode and electrolyte. Recently, electrochromic energy storage windows (EESWs) that integrate functions of energy storage and electrochromism are seen as a novel alternative to many existing solutions in the fields of self-powered displays and energy-efficient smart windows. Recently, colour-tuneable (nonemissive-red-yellow-green) self-powered EESW is reported by Zhai et al. [77], which is using the phenomenon of utilising Prussian blue (PB) as a controller of the fluorescent component of CdSe quantum dots. Many electrochromic-luminescent windows in different technologies were reported in the review by Kim et al. in 2019 [78].

In EESW, chemical reactions allow the device to collect the energy during the colouring phase (charge), e.g., during the day and discharge while bleaching. The energy stored in the ECCs could be used during the night for powering, e.g., the LED lighting, but could be also used to switch their optical modulation. The materials used in the class of electrochromic devices are mainly $WO_3$, $MnO_2$, and NiO, but the possibilities are wider. This type of EC device does not require external voltage to trigger the bleaching process, as the stored energy could be used to perform this task, as previously proved by Nguyen [73].

Electrochromic energy storage windows (EESW) have recently become the focus of many research projects that produced different results. In 2019, Cao et al. [79] reported a dual-band electrochromic energy storage window capable of independent control of the NIR and visible light at the dynamic range of VIS and NIR light modulation (89.1% at 550 nm and 81.4% at 1600 nm). The device is simultaneously capable of a high charge-storage capacity of 466.5 mAh m$^{-2}$ at 150 mA m$^{-2}$ of current density. The technology used was Ta-doped $TiO_2$ nanocrystals used for the working electrode (WE). In the year 2020, Wang et al. [55] presented an EESW based on Prussian blue that presented an energy storage ability of average output voltage of 1.24 V and an aerial capacity of 78.9 mAh m$^{-2}$. In this case, a battery-type Prussian blue (PB, $Fe_4{}^{III}[Fe^{II}(CN)_6]_3$)/Zn EC window was demonstrated with the remarkable transmittance modulation of $\Delta T_{vis}$ = 84.9% at 633 nm. In 2018, Kim et al. [80] reported the electrochromic capacitive windows (ECCs) with high transparency (>72%) with the use of blue (PR-Br) and red (Th-OR) EC polymers combined in a single device. Thin polyaniline (25 nm thick PANI) film was used as a capacitive layer. The energy stored in the blue ECCs could be transferred to charge the red ECC or to light a LED. The laboratory-scale device that performed with the power density of 58.8 kW kg$^{-1}$.

EESW is exhibiting typical problems for all EC devices. Practically all reported EESW devices are blue or green in the coloured state and are not completely transparent in the bleached state. In 2019, Xie et al. [81] demonstrated EESW with transparent-to-dark electrochromic with the use of Mo-doped tungsten oxide ($WO_3$) and nanoflake $MnO_2$ film electrode, which exhibited an aerial capacitance of 19.1 mF cm$^{-2}$ and presented the optical modulations of over $\Delta T_{vis}$ = 60% in all visible bands (420–800 nm). In the same paper, Xie et al. presented the comparison of EESW materials, stating that the best performance is currently achieved with the use of $WO_3$/$WO_3$ electrodes that demonstrate the areal capacitance of 22.0 mF cm$^{-2}$. In 2018, Wang et al. [82] reported mesoporous $WO_3$ film on FTO glass via a facile dip-coating sol-gel method. Mesoporous $WO_3$ film exhibited advantages including high transparency, good adhesion, and high porosity. The device in the laboratory scale showed noticeable electrochromic energy storage with a specific capacity of 75.3 mAh g$^{-1}$. In May 2020, Pan et al. [83] demonstrated a device using novel NiO/PB composite hexagon nanosheets that exhibited a high areal capacitance of 11.50 mF cm$^{-2}$. NiO/PB composite nanosheet electrode exhibited higher exchanged charge density, larger optical modulation, and better cyclic stability. An additional advantage of the presented solution was the large optical modulation $\Delta T_{vis}$ = 67.6% at 630 nm, and the fact that the process of charging/discharging can be repeated for at least 4000 cycles with little decay. For a comparison of the results, see Table 5.

**Table 5.** The comparison of the performance of electrochromic energy storage windows, including the technologies, where the data are available.

| No. | Team | Year | Type/Technology | $\Delta T_{vis}/\Delta T_{NIR}$ | Energy Storage (W, mAh, Areal Capacitance) |
|---|---|---|---|---|---|
| 1 | Sheng et al. | 2019 | Ta-doped $TiO_2$ nanocrystals | 89%/81% | 466.5 mAh m$^{-2}$ |
| 2 | Wang et al. | 2020 | Prussian blue | 84.9%/n/a | 78.9 mAh m$^{-2}$ |
| 3 | Kim et al. | 2018 | blue and a red colour ECP | n/a | 58.8 kW kg$^{-1}$ |
| 4 | Xie et al. | 2019 | Mo-doped $WO_3$ | 60%/n/a | 19.1 mF cm$^{-2}$ |
| 5 | Wang et al. | 2018 | mesoporous $WO_3$ | 75.6%/n/a | 75.3 mAh g$^{-1}$ |
| 6 | Pan et al. | 2020 | NiO/PB composite nanosheets | 67.6%/n/a | 11.50 mF cm$^{-2}$ |

### 5.4.5. Hybrid EC/TC Solutions

Electrochromic devices (EC) can be integrated with thermochromic (TC) devices into a single apparatus. The most frequently used method employs ultra-thin layers of tungsten trioxide ($WO_3$) for EC performance and vanadium oxide for $VO_2$-based thermochromic cells. Such a hybrid device can control optical and solar energy transmission independently as a response to electric current and the change in temperature. The biggest advantage is that this technology allows for selective modulation of VIS and NIR wavelength ranges. In 2019, the team of Lee et al. [27] reported the EC/TC system that integrated two technologies into one all-solid-state device using a solid electrolyte (tantalum oxide $Ta_2O_5$). TC layer $VO_2$ was deposited directly on the ITO positive electrode. The device exhibited four states. With the applied voltage ($-2$V), colour changed from transparent to deep blue due to the EC reaction. Then, the coloured device was heated (80 °C) and the device became even darker because of the additional TC behaviour further decreasing the transmission. Then, the voltage was inverted ($+2$V), and the transmittance was increased (as of the EC phenomenon). Finally, the device was cooled and returned to its original state. The $\Delta T_{vis}$ between transmittances in the bleached and coloured state was decreased by approx. 30% due to the synergic effect of EC and TC layers (e.g., from $T_{bleached}$ = 73.57% in room temperature to $T_{coloured}$ = 8.33% in 80 °C). For a comparison of the results, see Table 6 and Supplementary Materials.

**Table 6.** The performance of Hybrid EC/TC solutions including the technologies, where the data are available.

| No. | Team | Year | Type/Technology | $\Delta T_{sol}/\Delta T_{vis}$ | Remarks |
|---|---|---|---|---|---|
| 1 | Lee et al. | 2019 | tantalum oxide $Ta_2O_5$ | 45%/45% | at 20° |
| | | | | 36%/34% | at 80° |

### 5.4.6. EC + OLED Lighting

Sold state electrochromic cell could be also coupled with a solid-state organic light-emitting diode (OLED) producing the device capable of regulating the transparency and simultaneously providing artificial light. This type of device was demonstrated for the first time by the team of Lu et al. in 2018 with the use of PEDOT polymer for tuning the light-emitting direction. The colour of the glass changes from pale blue to dark blue upon the application of an external voltage. Therefore, "the utilization of this coloration or discolouration of the glass substrate can adjust the light-emitting direction of OLED" [84]. When the EC component of the device is switched to semi-transparent mode, the device exhibits a "graceful spatial impression"; when the glass is switched to a highly opaque mode, "top emission with a significantly enhanced luminance efficiency" is visible. Furthermore, in 2018, Cossari et al. [85] presented the device named ECOLED capable of tuning the transmittance (EC phenomenon) and producing light by electroluminescence,

simultaneously or independently. The architecture of the device was based on 300 nm thick $WO_3$ layers and showed blue tinting via reduction of $W^{6+}$ sites to $W^{5+}$. ECOLED device is possible to be applied in many fields: in EC smart windows, EC mirror, and technologies of transparent displays. The OLED component exhibits luminance above the minimum values required for display and lighting applications, which are 300 cd m$^{-2}$ and above 800 cd m$^{-2}$, respectively. The technology also allows for the simultaneous control of light transmittance and artificial lighting achieving an optical contrast of $\Delta T_{vis}$ = 57% at 650 nm and high colouration efficiency. For a comparison of the results, see Table 7.

**Table 7.** The performance of EC + OLED lighting solutions including the technologies, where the data are available.

| No. | Team | Year | Type/Technology | $\Delta T_{vis}/\Delta T_{NIR}$ | Remarks |
|---|---|---|---|---|---|
| 1 | Lu et al. | 2018 | PEDOT polymer | 25%/n/a | switch between two semi-transparent states, $\approx$45% $\approx$70% absorption, light emitting 35.0 and 7.5 cdA$^{-1}$ |
| 2 | Cossari et al. | 2018 | | 57%/n/a | luminance from 300 cd m$^{-2}$ to 800 cd m$^{-2}$ |

### 5.4.7. EC Devices Powered by Solar Cells (DSSC-EC)

An EC device could be combined with dye-sensitised solar cell (DSSC), resulting in the device named DSSC-EC. DSSCs are solar cells of low construction cost belonging to the group of thin-film solar cells. The first dye-sensitised solar cell coupled with an electrochromic layer (EC-DSSC) was reported by Bechinger et al. [86] in 1996. The DSSC-EC devices were gradually improved. Wu et al. [87] report the device manufactured with the use of PProDOT-Et$_2$ that plays two roles, namely, as the electrochromic thin layer for electrochromic action and the counter electrode for dye-sensitised solar cell (DSSC). Under the light illumination, the photoactive layer of a dye-adsorbed $TiO_2$ is sensitised initially and produces electrons that cannot migrate to the electrochromic layer, as the circuit is open. When the circuit is closed, electrons generated from the dye-sensitised $TiO_2$ can migrate through the gel electrolyte, and the device is darkened. In this case, the transmittance can be changed reversibly from $T_{VIS(bleached)}$ = 46% to $T_{VIS(coloured)}$ = 15% at 590 nm, which is triggered without any external voltage. In 2019, Costa et al. [88] reported a system that is self-powered and changes colour spontaneously when illuminated. The device exploits the architecture of dye, $TiO_2$, and $WO_3$ as an electrochromic layer in the different configurations. The performance of the device changes according to the location of the thin layer of $WO_3$. When the $WO_3$ is deposited on the working electrode, the device exhibits the best colour contrast; when it is applied on photoanode (anode of the photoelectric cell), the electricity conversion efficiency of ca. 7% was obtained in the closed-circuit conditions.

### 5.5. Application of Nanostructures in EC Device Design

Nanostructured electrochromic materials are used in electrochromic devices to achieve increased colouration efficiency, faster-switching speed, and longer cycling lifetime [78]. The mechanism behind the increased efficacy is that a nanostructure has a large specific surface area and high aspect ratio that allows for the better penetration of the electrolyte and to improve the ion diffusion [15].

It must be said that electrochromic devices based on bulk $WO_3$ usually demonstrate a slower switching time. To overcome this drawback, one-dimensional (1D) nanostructured materials, such as nanowires and nanotubes, have been used and tested by different teams since the beginning of the 20th century, as covered in the review from the year 2010 by Wang et al. [89]. Recently, many tungsten oxide ($WO_3$) nanostructures including nanowires [90,91], nanotubes [92], nanobelts [93], nanorods [94,95], and spindle-shaped $WO_3$ [96] have been synthesised.

In the year 2012, $TiO_2$ nanowires were proved to demonstrate enhanced optical transparency in the visible range by Chen et al. [97] and Tokudome et al. [98]. Regarding

reports from the years 2015–2020, in 2017, Lu et al. [99] presented a device using the arrays of tungsten oxide nanorods with a diameter of 22 nm and length of 240 nm. This architecture provided high surface area, uniform thickness, and good adhesion to the substrate. The produced EC device demonstrated optical modulation of $\Delta T_{vis}$ = 41.2%, at 632.8 nm under the voltage of 0.1 V for 10 s.

In 2018, the team of Najafi-Ashtiani et al. [100] demonstrated an EC device featuring synthesised Ag nanorods that are covered by tungsten oxide ($WO_3$) shells. The device exhibited a significant optical modulation of 36.81% at 633 nm, and a relatively fast switching time of 5.7 s.

In 2019, the team of Shi et al. [101] presented a hybrid device comprising hybrid nanorods are composed of $WO_3$ nanocores wrapped by thin amorphous PEDOT nanoshells. Wrapped $WO_3$ nanocores present EC behaviour of a much shorter response time of approx. 3.7 s. than bare $WO_3$ nanocores. The dynamic analysis presented by the team of Shi et al. suggests a synergistic effect between the $WO_3$ nanocore and the PEDOT nanoshell. As a result, the colour depth and optical contrast of the hybrid nanorods can be modulated by adjusting the applied voltage and the deposition of the PEDOT nanoshell.

Consequently, in 2020, the team of Shi et al. [102] presented a bilayer hybrid $WO_3$ nanoarray device composed of crystalline $WO_3$ nanobowls. The hybrid device exhibits very good electrochromic performance in both visible and NIR wavelength ranges of colour contrast, $\Delta T_{vis}$ = 93.9% at 633 nm and $\Delta T_{NIR}$ = 89.6% at 1500 nm, respectively. The nanobowls are produced with the use of polystyrene (PS) spheres template with a diameter of 500 nm. The $WO_3$ layer was deposited on the spheres and then the spheres were removed by solving them in a solvent. The resulting surface was comprised of the nanobowls of $WO_3$.

Mesoporous material is a material containing pores with diameters between 2 and 50 nm, according to IUPAC nomenclature. Mesoporous $WO_3$ film usually exhibits improved optical modulation performance. Optical modulation up to 71% at the wavelength of 633 nm was presented by Wang et al. in a previously cited report [82]. The mesoporous structure exhibits a noticeable electrochromic energy storage performance with a large optical modulation up to 75.6% at 633 nm. For a comparison of the results, see Table 8.

**Table 8.** The performance of EC with the use of nanostructures including the technologies, where the data are available.

| No. | Team | Year | Type/Technology | $\Delta T_{vis}/\Delta T_{NIR}$ | Remarks |
|-----|------|------|-----------------|------------------------------|---------|
| 1 | Lu et al. | 2016 | tungsten oxide nanorods | 41%/n/a | at 632.8 nm |
| 2 | Shi et al. | 2018 | $WO_3$/PEDOT core/shell hybrid nanorod arrays | 80%/n/a | only $WO_3$ |
| | | | | 26%/n/a | only PEDOT |
| | | | | 72%/n/a | $WO_3$ and PEDOT |
| 3 | Najafi-Ashtiani et al. | 2018 | Ag nanorods | 37%/n/a | range 32–37% |
| 4 | Shi et al. | 2020 | $WO_3$ nanoarray | 94%/90% | areal capacitance 47.4 mF/cm$^2$ |
| 5 | Wang et al. | 2018 | mesoporous $WO_3$ | 75.6%/n/a | capacity of 75.3 mA h g$^{-1}$ |

## 6. Discussion—Main Challenges

The most recent examples of ECD are presented in the paper also for some speculation considering the most important areas of future application. Most of them are currently in the development phase at the stage of working on a small-size prototype. In most cases, the prototypes are only a few square centimetres of surface area, and the manufacturing technology is available only on a laboratory scale. It must also be clearly stated that no unified standard of the results presentations is valid/present, and therefore, different teams use different metrics. This might limit the accuracy of the evidence and results included in the review. Although it is needed to exercise caution in interpreting these presented data

because of the limited number of reviewed papers (105), these findings nonetheless appear to be largely in line with systematic reviews by other researchers [12,13].

Nevertheless, the future application could be briefly discussed as the aspect of the technology that should be focused on, to bring a wider range of opportunities.

The optical performance of electrochromic smart windows should be improved, especially for large-scale commercial applications. The main issues are low optical contrast and long response time, as addressed by Zhang et al. [103]. Wider customer acceptance for electrochromic systems is prevented by the still-to-be-solved drawbacks of the EC systems, despite the long-time research, development, patents, and start-ups [18]. The main disadvantage of electrochromic materials is the fact that they are unable to produce neutral colouration, and the neutral grey electrochromism is in the development stage. Electrochromic devices based on the tungsten trioxide ($WO_3$) switch from deep blue to transparent, while the nickel oxide (NiO) switches from transparent to brown. This colour filtering is generally considered as a disadvantage, as the light filtered through the smart glass influences the colour perception in the room [104]. In addition, blue light is considered to be a dangerous part of the visible spectrum because of the generation of reactive oxygen species in the retina [105].

Another disadvantage of current EC technologies is the long switching time. This also should be improved to earn more customer acceptance of the technology. Small samples of electrochromic materials in the conditions of laboratory experiment present fast coloured/bleached transitions that could be measured in seconds. However, window-size (approx. 2.5 sq. m), commercially available solutions take much more time to switch (up to 20 min) due to the increased resistance of the system and a greater amount of electrical charge that must be transmitted through electrical contacts and electrolyte [26]. This has been considered both as an advantage—a desirable feature that permits the eye to light-adapt [106]—but also as an evident disadvantage, preventing quick reaction, e.g., to moving clouds in the sky. ECD smart windows also lack long-term cyclic stability [107] and feature low optical contrast [61].

Another drawback of the currently available commercial solution is the high capital cost, averaging 540–1080 USD/m$^2$ [35].

## 7. Conclusions

The presented review systematically summarises the recent progress on prototype smart windows solutions that are currently in the stage of research and examines the quantitative parameters of smart window devices. The previous section focused on the main challenges that the technology is facing, although the conducted review allows the following conclusions to be formulated. The key findings are summarised by the following categories:

### 7.1. Smart Windows

- The smart window is a mature technology that has been studied for many years in many variations. The proof of this is the industrial application of the selected technologies and the presence of brands (e.g., View, Sage, Gesimat, Gentex, ChromoGenics);
- Smart windows have not achieved significant market penetration due to the factors discussed in the previous paragraph;
- Smart windows must also switch deeply and quickly enough to mitigate glare and prevent user discomfort, or they will not gain user acceptance;
- In the case of so-called privacy windows, the haze effect must be considered when discussing the transparency/translucency (high) and energy-saving performance (relatively low);
- The review shows that the widespread adoption of smart window technology calls for better performance and cost competitiveness.

### 7.2. Electrochromics

- Many ECD technologies are currently at the stage of research and development as multicolour, neutral black, spectrally selective, energy storage, and generation;
- Hybrid technologies are of special attention, e.g., smart glass joining in one device the PV and ECD (self-powering smart windows), or ECD and OLED light-emitting diodes or ECD and TC technologies;
- Hybrid devices usually expose lower either VIS or NIR performance while presenting other functionalities;
- Within the review, the best visual performance was presented with the use of two technologies: ECD featuring Ta (tantalum) doped titanium oxide ($TiO_2$) with the modulation $\Delta T_{vis}$ of 86.3% [75], and $WO_3$ nanoarray with the impressive modulation $\Delta T_{vis}$ of 93.9% [102];
- Dual-band (NIR/VIS) technologies are promising in the context of energy flow management. In this category, the best performance was achieved by a device which is capable of shielding 96.2% of the NIR irradiation from 800 to 2500 nm [71];
- The application of nano-technologies seems to be opening a wealth of new opportunities presenting the best performance;
- The observed tendency is that the complexity of ECDs is growing, especially the number of layers, the architecture of nano-structures, and manufacturing technology (deposition sequence).

The results provide important insight into the determination of the most promising technologies. The present review also demonstrates the need to standardise the metrics that are used to make it easier and faster to benchmark different technologies. Although the most common metric reported by the authors was luminous/visual transmittance $T_{vis}$, the used metrics do not always reflect the most important parameters within the described category. Moreover, the relationship between the performance and adopted technology could be recognised in the paper.

### 7.3. Limitations of the Study

Limitations of the review study usually result from the insufficient number of included publications. To counteract possible bias, PRISMA protocols were used in the writing of this paper, but this potential limitation should nevertheless be explicitly mentioned. The limitations are due to (i) different metrics used by different authors (e.g., only modulation values were given without the $T_{coloured}$ and $T_{bleached}$ values); (ii) values of $T_{coloured}$ and $T_{bleached}$ were not explicitly stated in papers and had to be retrieved from the graphs; and (iii) the small size of laboratory samples (possibly not replicable results on a larger scale).

Despite the constraints, the study, the method, and the results are original and valuable contributions to the review of the most recent electrochromic technologies.

### 7.4. Future Application

Despite the challenges that ECD technology is currently facing, there are also opportunities strongly connected with ECD application in the built environment. Dual-band electrochromic materials would finally allow for the regulation of daylight and heat flow into buildings. Both spectral ranges (VIS and NIR) are particularly interesting. In future practice, the separate, independent regulation of VIS and NIR radiation would be very beneficial for passive homes. ECD in the coloured state is used to optimise a building's energy performance in the summer by reducing the heat load; this results from the fact that energy savings with the use of ECD are "due to a lower cooling demand" [10]. Especially, coupled EC/TC devices might show additional benefits here. However, it must be remembered that NIR radiation fulfils an important function in the building's energy balance in winter, as the greenhouse effect is used in the winter season to raise the internal temperature. A device that can independently screen both wavelength ranges would be beneficial for the overall building's energy balance. The use of ECD shows benefits not

only in the energy context but also in the reduction of the carbon footprint of new and existing office buildings [108].

Multicolour ECD solutions might be also used to regulate internal rooms' lighting atmosphere and compensate for the natural daylight. Multicolour ECD might be used as a large-scale transparent display. Simultaneously, neutral black electrochromism is widely sought after as the most correct solution for light dimming in work and living spaces, preventing glare and thermal discomfort without impairing the glazing's primary function of direct eye contact with the surrounding [106].

## 8. Summary

Buildings account for 40% of total energy consumption in the European Union and even more in other countries whose energy-saving policies are not so up to date. To help bring about a carbon-neutral future, it is thus necessary to reduce the consumption of energy from non-renewable sources and decrease the energy demand through energy-saving, environmentally friendly technologies. In this perspective, ECD, which can regulate the flow of energy during the day and provide artificial lighting during the night, might reduce the demand for separate lighting systems and reduce the demand for energy for cooling. Currently existing and still developing ECD technologies open up a very wide range of possibilities for their use in construction. This is the reason why broad research is necessary, and hopefully, it will finally be able to identify technologies that have the potential of scaling, which seems to be one of the most important challenges of the ECD technology.

**Supplementary Materials:** The following are available online at https://www.mdpi.com/article/10.3390/su13179604/s1, Table 9: All the systems described in the paper in a tabularise form, with the data where available. Figure. X Prisma checklist, Figure. X Prisma flow diagram.

**Funding:** The financial support of this study that was provided through a grant entitled: "New trends in architecture of transparent facades—formal experiments, technological innovations", ref. no. 2014/15/B/ST8/00191 by the National Science Centre, Poland. The APC was funded by Wroclaw University of Science and Technology.

**Institutional Review Board Statement:** Not applicable.

**Informed Consent Statement:** Not applicable.

**Data Availability Statement:** Prisma checklist is available at https://drive.google.com/file/d/1erEaqla-gXLLsYf3alorAmPcU4MIorII/view?usp=sharing (accessed on 20 August 2021), Prisma flow diagram is available at: https://drive.google.com/file/d/1L5pnUIrEqxkVXTp5UDEEUsEWf64iKEFg/view?usp=sharing (accessed on 20 August 2021).

**Conflicts of Interest:** The author declares no conflict of interest.

## Nomenclature

| | | | |
|---|---|---|---|
| TC | thermochromic | DSSC | dye-sensitised solar cell |
| EC | electrochromic | OLED | organic light-emitting diode |
| GC | gasochromic | MEMS | microelectromechanical systems |
| TC | thermochromic | EESW | electrochromic energy storage windows |
| ECD | electrochromic device | PDLC | polymer dispersed liquid crystal |
| $T_{vis}/T_{lum}$ | visible transmittance | SPD | suspended particle devices |
| $T_{sol}$ | solar transmittance | PEDOT | conducting polymer based on 3,4-ethylene dioxythiophene |
| $T_{NIR}$ | near-infrared transmittance | UV | ultraviolet |
| $\Delta T_{vis}$ | visual modulation | TRL | technology readiness level |
| $\Delta T_{sol}$ | solar modulation | | |
| $\Delta T_{NIR}$ | near-infrared modulation | | |

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
