# Peer review of "A Systematic Review of the Most Recent Concepts in Smart Windows Technologies with a Focus on Electrochromics"

_sustainability, doi:10.3390/su13179604_

Round 1

Reviewer 1 Report

The topic is attractive.
It might be an effective support for the design focused on low and carbon energy buildings. Nevertheless, a topical aspect that supports all evaluation is not adequately explained, and I mean the method that defines the formula (1)
ΔT = T(coloured) - T(bleached) [%] (1).
Please add references
In addition, explain better the process that is adopted in described technologies

Reviewer 2 Report

Thank you for submitting your paper “A systematic review of the most recent concepts in smart windows technologies with a focus on electrochromics.” to the Journal of Sustainability.
In my opinion, it is a valuable and interesting article. Overall, it is well written but needs some improvements before publication. 
The originality of the paper needs to be stated clearly and the most significant results.
In general, I strongly suggest clearly explaining the significant findings and the real novelty of the work, implementing the conclusion part.
Other studies focus on this issue:
[https://doi.org/10.1016/j.enbuild.2021.111184] conducted a critical review of fenestration/window system design methods for high-performance buildings. 
[https://doi.org/10.1016/j.apenergy.2019.113522] conducted a critical review on thermochromic smart window technologies for building applications.
[https://doi.org/10.1016/j.energy.2016.06.002] stated that the windows represent approximately 30–50% of transmission losses through the envelope and used a CFD model to evaluate the thermal performance of window frame profiles. 
In Figure 1, color necessary?
Please, add the table of nomenclature with technical data and all acronyms used.
